# Immunomodulatory Properties of Immune Checkpoint Inhibitors—More than Boosting T-Cell Responses?

**DOI:** 10.3390/cancers14071710

**Published:** 2022-03-28

**Authors:** Michael Kuske, Maximilian Haist, Thomas Jung, Stephan Grabbe, Matthias Bros

**Affiliations:** Department of Dermatology, University Medical Center Mainz, Langenbeckstraße 1, 55131 Mainz, Germany; mikuske@uni-mainz.de (M.K.); mhaist@uni-mainz.de (M.H.); thjung@students.uni-mainz.de (T.J.); stephan.grabbe@unimedizin-mainz.de (S.G.)

**Keywords:** immune checkpoint inhibitors, CTLA-4, PD-1, PD-L1, immunotherapy, immunological effects, metastatic melanoma, tumor microenvironment, emerging immune checkpoints

## Abstract

**Simple Summary:**

Tumor immune evasion is mediated in large part by the inhibition of anti-tumor T cell responses. Both the induction of tumor antigen-specific T cells and the activation state of T effector cells may be attenuated by surface receptors, which upon binding to counter receptors on immunoregulatory cell types and tumor cells, induce inhibitory T cell signaling. Immune checkpoint inhibitors (ICI) are antibodies that block interaction of these receptor pairs and thereby prevent T cell inhibition. Only a small percentage of tumor patients are responsive to treatment with ICI, which on the one hand raises the issue of possible reasons for failure, and on the other hand has spurred the development of additional ICI targeting other T cell inhibitory receptors. Our review aims to summarize knowledge on the functional role of these (inhibitory) receptors by additional types of leukocytes, and consequences of receptor blockade by ICI as a potential cause for unwanted side effects limiting the success of therapy. Deeper knowledge in this regard is a prerequisite for the development of more refined combination therapies.

**Abstract:**

The approval of immune checkpoint inhibitors (ICI) that serve to enhance effector T-cell anti-tumor responses has strongly improved success rates in the treatment of metastatic melanoma and other tumor types. The currently approved ICI constitute monoclonal antibodies blocking cytotoxic T-lymphocyte-associated protein (CTLA)-4 and anti-programmed cell death (PD)-1. By this, the T-cell-inhibitory CTLA-4/CD80/86 and PD-1/PD-1L/2L signaling axes are inhibited. This leads to sustained effector T-cell activity and circumvents the immune evasion of tumor cells, which frequently upregulate PD-L1 expression and modulate immune checkpoint molecule expression on leukocytes. As a result, profound clinical responses are observed in 40–60% of metastatic melanoma patients. Despite the pivotal role of T effector cells for triggering anti-tumor immunity, mounting evidence indicates that ICI efficacy may also be attributable to other cell types than T effector cells. In particular, emerging research has shown that ICI also impacts innate immune cells, such as myeloid cells, natural killer cells and innate lymphoid cells, which may amplify tumoricidal functions beyond triggering T effector cells, and thus improves clinical efficacy. Effects of ICI on non-T cells may additionally explain, in part, the character and extent of adverse effects associated with treatment. Deeper knowledge of these effects is required to further develop ICI treatment in terms of responsiveness of patients to treatment, to overcome resistance to ICI and to alleviate adverse effects. In this review we give an overview into the currently known immunomodulatory effects of ICI treatment in immune cell types other than the T cell compartment.

## 1. Introduction

Until a few years ago melanoma patients with distant metastases had poor prognoses, with median survival rates found between six and twelve months [1]. The approval of the checkpoint inhibitor (ICI) ipilimumab, which targets the immunoinhibitory receptor cytotoxic T lymphocyte associated protein (CTLA-)4 for the treatment of metastatic melanoma, more than a decade ago initiated an era of considerable progress in melanoma treatment [2]. The additional approval of other types of ICI targeting the PD-L1/PD-1-inhibitory axis [3] as well as the introduction of mitogen-activated protein kinase (MAPK)-targeting inhibitors [4] has strongly improved prognosis in the majority of melanoma patients to date [5,6] and these types are also employed for the treatment of other types of tumors.

This review aims to summarize mechanisms of tumor immune evasion, which are mediated in large part via the upregulation of immune-inhibitory receptors by tumor cells, termed immune checkpoints (IC), and by immune cells that are reprogrammed by tumor-derived soluble mediators. Due to the high mutational load found in melanoma and its strong immunogenicity, malignant melanoma is considered a model tumor for immunotherapy. Hence, we discuss the different mechanisms of immune evasion mostly with a focus on the example of metastatic melanoma. Further, we describe the mode of action of ICI aimed at counteracting the anti-inflammatory effects of IC, which in the case of anti-CTLA-4 and anti-PD-1 have been granted early approval for melanoma treatment.

In the main part of the review, we discuss the role of ICI-based therapies in enhancing non-CD8^+^ T-cell-based immune responses within cancer. In particular, we will outline the expression of CTLA-4 and the PD-1/PD-L1 proteins in non-CD8^+^ T-cell compartments, such as CD4^+^ T cells, innate lymphoid cells, B cells and myeloid cells and describe the role of ICI for these immune cell populations. By understanding the immunological effects of ICI beyond the expansion of T effector cells, this review may help to explain in part both the current limitations in ICI efficacy and the character of immune-related adverse effects (irAEs). Lastly, we briefly elaborate on other ICs that are currently being monitored for efficacy in clinical trials.

## 2. Characteristics of Malignant Melanoma as a Model Tumor in Immune-Oncology

Melanocytes originate from the neural crest, and are found mainly in the basal layers of the epidermis, in hair follicles, along mucosal surfaces, in the meninges, and in the choroid of the eye [7]. Skin keratinocytes produce melanocyte-stimulating hormone in response to UV-induced DNA damage [8], finally resulting in the production and release of melanin by melanocytes, which is able to scatter and absorb UV radiation [9]. However, UV radiation is the most frequent cause of melanoma-associated mutations, which is reflected by the frequent observation of UV-associated mutation signatures in melanoma lesions [10]. Malignant melanoma displays one of the highest genomic mutation loads among all solid tumors, resulting in the expression of various tumor-associated antigens (TAA) that constitute potential targets for anti-tumor immune responses [11]. In addition, overexpression of cell-line-specific or so-called “cancer/testis” antigens in melanoma, such as the differentiation antigens gp100, Melan-A or the highly immunogenic MART/MAGE antigens, were among the first cancer antigens to be identified and classified, thus allowing researchers to develop TAA-directed immunotherapies in melanoma [12,13,14].

In addition to tumor mutations that affect cell proliferation and survival largely via constitutive activation of the MAPK [15] and the phosphoinositide 3-kinases (PI3K) [16] pathways, as well as inhibition of tumor suppressors, such as phosphatase and tensin homolog (PTEN) [17], which impairs PI3K signaling [18], mechanisms that promote tumor invasion and infiltration are of particular importance in melanoma [19]. Observations in melanoma have provided proof of principle that the immune system is capable of mounting an anti-tumor immune response, as visually confirmed by partial tumor regression or immune infiltration in primary melanomas, as well as by spontaneous or treatment-related vitiligo [14]. As a result, immunotherapies in melanoma have been at the forefront of immuno-oncology research.

## 3. Tumor-Induced Immune Modulation

Besides genetic changes, interactions within the tumor microenvironment (TME) and modulations of the immune response also promote tumor development [20]. Here, it has been shown that ICs that limit T-cell activation contribute essentially to tumor resistance [21]. Tumors are able to exploit this mechanism by expressing specific immune checkpoints, especially ligands of the T-cell receptor PD-1, PD-1 ligand (PD-L1) and PD-L2, which by engagement of PD-1 on T cells prevent their activation [22].

Further, it has been demonstrated that tumors suppress an anti-tumor immune response by impairment of major histocompatibility complex (MHC)-I expression, which limits the presentation of TAA to cytotoxic T lymphocytes (CTL) [23]. Next, tumors secrete immunomodulatory cytokines, such as vascular endothelial growth factor (VEGF) as well as other mediators, such as nitric oxide and prostaglandins, which contribute to chronic inflammation within the TME and tumor progression [20]. The chronic inflammatory state generated via the accumulation of cytokines and growth factors promotes the recruitment and induction of immunosuppressive cells, such as regulatory T cells (Tregs), myeloid-derived suppressor cells (MDSCs) and tumor-associated macrophages (TAMs) [20]. MDSCs are a heterogeneous group of immature granulocytic and monocytic cells that are induced and activated by TME-derived factors and confer immunosuppressive activities [24]. Of note, granulocytic MDSCs and polymorphonuclear granulocytes (PMNs) display a strongly overlapping immunophenotype and can be distinguished only by gradual differences in marker expression [25] and their functional activity. TAMs are derived from tumor-infiltrating monocytes, which acquire an M2-like immunophenotype within the TME [26]. Immunosuppressive cells of myeloid origin also express checkpoint receptors, which upon engagement with their ligands on T cells inhibit the latter [20].

In this regard, tumor-derived immunoregulatory mediators, such as IL-6, IL-10, TGF-β, CC-chemokine-ligand-2, VEGF and granulocyte-monocyte-colony-stimulating-factor (GM-CSF), as well as prostaglandin E2, metabolic dysregulation and tissue hypoxia, contribute to the polarization of monocytes and macrophages towards TAMs [27]. TAMs secrete proteases, and cytokines, such as epidermal growth factor (EGF) or VEGF, which may further promote the epithelial-mesenchymal transition of melanoma cells, thus favoring cancer cell invasion, metastasis and neoangiogenesis [28].

In addition, tumors promote the induction of Tregs [29] and redirect the differentiation of myeloid progenitor cells towards MDSCs [30] via immunomodulatory mediators, such as IL-10. MDSCs and Tregs, which are mobilized by cytokines, such as TGF-β and C-X-C motif chemokine ligand 5, suppress the immune response by various mechanisms, such as inhibition of antigen presentation by dendritic cells (DCs), activation of T cells and B cells, and NK cell cytotoxicity [31]. MDSCs and Tregs interact both via soluble mediators via receptor interactions to exert their immune-regulatory properties [20].

Further, cancer-associated fibroblasts (CAF) as activated by factors produced by tumor and stromal cells (e.g., TGF-β, EGF, platelet-derived growth factor and fibroblast growth factor 2) [32], support tumor growth by secreting EGF, hepatocyte growth factor, insulin growth factor-like family member (IGFL)1 and IGFL2 [33], and promote angiogenesis via VEGF release [34]. It has been shown that the crosstalk between CAF and TAM sustains tumor progression [35] and that the generation of IL-1α and IL-1β by melanoma cells may promote this malignant interaction [36]. Moreover, Young and coworkers demonstrated that TAMs were also able to generate IL-1β, which induced the secretion of IL-8 and growth-regulated oncogene-α by CAF [37]. These mediators enhanced expression of B-Cell chronic lymphocytic leukemia/lymphoma (BCL)-2 via a C-X-C motif chemokinereceptor2-stimulating secretome in melanoma cells, contributing to their survival [27,38]. In accordance, melanomas displayed an attenuated growth rate when induced in IL-1 receptor-deficient mice, underpinning the pro-tumorigenic role of IL-1β.

Altogether, the TME constitutes an immunoregulatory environment, which impairs infiltration of activated immune cells, including T effector cells, which could exert anti-tumor activity at the vascular border [39]. Moreover, the TME interferes with the state of activation of infiltrating immune cells by various mechanisms, including soluble factors, hypoxia, and cell–cell interactions (see Figure 1) [20].

## 4. Treatment of Metastatic Melanoma and Other Solid Tumors with ICI

Most melanoma patients initially present in early clinical stages and are treated with surgical resection of the primary tumor with appropriate safety margins and sentinel node biopsy [40]. Until a decade ago, no effective systemic therapy was available for patients suffering from metastatic melanoma [41]. MAPK targeting therapies and the introduction of ICI, which counteract T-cell inhibition mediated by CD80/CD86-CTLA-4 and the PD-L1/PD-L2-PD-1-inhibitory signaling axes, prolonged the median overall survival (OS) of these patients to more than 5 years [42]. Next to metastatic melanoma, checkpoint blockade with ICI has also shown clinical efficacy in many other solid tumor entities [43,44,45]. Therefore, following the approval of ICI for the treatment of metastatic melanoma, checkpoint-inhibitors have in subsequent years been approved for other solid tumors, such as lung cancer, colorectal cancer (CRC), breast cancer and others. Although the main mode of action of ICI has widely been attributed to the disinhibition of CD8^+^ CTL, culminating evidence indicates that ICI may also confer clinical efficacy via additional immunological mechanisms. In the following, we summarize knowledge on the modes of action of CTLA-4 and PD-1/PD-(L1) targeting ICI on T cells, but also other types of immune cell types and tumor cells, which has been collected in preclinical studies using various murine tumor models.

### 4.1. Immunomodulatory Effects of Anti-CTLA-4 on Leukocytes and Tumor Cells

#### 4.1.1. Blockade of CTLA-4 Enhances the Generation of T Effector Cells

Tumor antigens are presented by antigen-presenting cells (APCs), such as DCs via MHC complexes, which bind to T-cell receptors displaying sufficient affinity for the MHC/antigen complex [46]. T-cell activation requires additional co-stimulation, which is conferred by costimulatory receptors of the APCs (e.g., CD80/CD86) that engage CD28 on the antigen-specific T cell. Naïve T cells express low levels of CTLA-4. Upon activation, CLTA-4 is induced in T cells to prevent excessive T-cell stimulation, while Tregs constitutively express CTLA-4 [47]. CTLA-4 has a much higher affinity than CD28 for CD80/CD86 binding, and in contrast to the homologous glycoprotein CD28, induces inhibitory signaling in the T cell [48]. Accordingly, CTLA-4 serves as an antagonist of CD28-mediated co-stimulation. The disruption of CD28 signaling is therefore generally accepted as the major mode of action conferred by CTLA-4. The importance of CTLA-4 to maintain self-tolerance is evident in CTLA-4 knockout mice, which die after only three to four weeks of age from exaggerated autoimmune responses, such as lymphoproliferative disorders [49].

Preclinical trials proved the therapeutic efficacy of CTLA-4-blocking antibodies by enhancing overall T-cell activation, yielding much stronger anti-tumor responses [50]. In accordance, in several clinical trials the anti-human CTLA-4 antibody ipilimumab demonstrated a substantial prolongation of progression-free survival (PFS) and OS (median OS: 10.1 months) in patients with metastatic stage melanoma [51,52,53]. Moreover, it has been observed that the application of ipilimumab may induce sustained anti-tumor responses, as documented by the occurrence of long-term survivors [53].

However, it has also been found that the application of ipilimumab frequently resulted in the occurrence of irAEs, with 15% being of grade 3–4, leading to treatment discontinuation [51,53]. The adverse toxicity profile encouraged the development of other ICIs, among which, PD-1 and PD-L1 inhibitors have been approved for the treatment of many advanced cancer diseases, as described in 4.2. ICI treatment in general, and combined immune checkpoint blockade with anti-CTLA-4 and anti-PD-1 antibodies in particular, is prone to cause irAEs due to interference with the overall tolerance-promoting role of the ICI targets [54]. IrAEs typically arise between 3 and 14 weeks from initial application of ICI, but may occur at any point during treatment or even after treatment discontinuation [55,56]. Most commonly, irAEs include the occurrence of gastrointestinal, hepatic, endocrinological or cutaneous symptoms, which may enforce an early discontinuation of treatment.

#### 4.1.2. Immune Effects of CTLA-4 Blockade beyond Direct Expansion of T Effector Cells

Although CTLA-4 has initially been described as a CTL-specific molecule, the main mechanisms of action described for anti-CTLA-4 antibodies are currently believed to be conferred by Tregs (Figure 2). Further, CTLA-4 blockade was reported to revert T cell exhaustion, thereby enhancing the cytotoxic effects of CTL [57]. However, these effects could not be uniformly confirmed in clinical studies, and the highest clinical activity of ipilimumab linked to Treg depletion has been observed for anti-CTLA-4 antibodies with high-affinity FcR polymorphism [58].

So far, studies have not been conducted with regard to whether ipilimumab directly counteracts Treg-conferred induction of the tryptophan-degrading intracellular enzyme indoleamine-2,3-dioxygenase (IDO) in DCs (see below). Interestingly, however, constitutive IDO expression by tumors was identified as a tumor immune-evasion mechanism [59], which researchers aimed to overcome by co-administration of ICI plus an IDO inhibitor [60]. However, clinical trials that have tested this combination in melanoma patients have so far failed to generate additional benefit for patients [61,62].

Concerning the finding of reverted CTL exhaustion by CTLA-4 blockade [57], it has also been suggested that anti-CTLA-4 antibodies may mediate their clinical effects, as anti-CTLA-4 antibodies attach to Tregs and thus contribute to Treg depletion via antibody-dependent cell-mediated cytotoxicity (ADCC) [63]. This effect was convincingly shown in mice, where anti-CTLA-4 antibody caused strong depletion of Tregs from the tumor [64]. In humans, the number of Tregs did not change in different cancer types after anti-CTLA-4 immunotherapy [65]. Hence, the impact of ADCC on anti-CTLA-4 efficacy could not be uniformly shown in clinical studies, yet the highest clinical activity of ipilimumab linked to Treg depletion has been observed for study participants with high-affinity FcR polymorphisms [58]. Anti-CTLA-4 antibodies with ADCC-enhanced FC parts are therefore in development.

Notably, in an in vitro study, peripheral blood mononuclear cells of healthy donors that were incubated with tetanus toxoid displayed impaired CD4^+^ T cell activation when ipilimumab was co-applied, due to binding of the constant IgG1 Fc part of the anti-CTLA-4 antibody to FcγRIII [66]. Assessment was not conducted with regard to which FcγRIII-expressing cell type was responsible for impaired T cell activation. To our knowledge, there has as yet been no assessment with regard to whether the IgG1 Fc part of ipilimumab (as well as other therapeutic antibodies) limits its anti-tumor effects in patients.

##### Myeloid Cells

CTLA-4 is not only expressed by T cells, but by numerous other leukocyte populations, and has been found to exert various immunological functions in a cell-type-dependent manner [67]. In this regard, the role of CTLA-4 blockade for regulatory myeloid cells, which act as another important immunosuppressive immune-cell population besides Tregs, has been in the focus of investigation more recently [67]. Here, it has been shown that CTLA-4 blockade in a PTEN knock-out mouse model of head and neck squamous cell carcinoma reduced the overall number of MDSCs and M2 macrophages, and accordingly decelerated tumor cell growth [68]. These findings indicate that CTLA-4 may regulate the induction and functional state of MDSCs and macrophages [69]. In accordance, it has been observed that CD11b^+^CTLA-4^+^ cells mediated tumor progression via IL-1β production and by CTLA-4 signaling in a mouse model of CRC [70]. More specifically, Yang and coworkers reported that MDSC-mediated immunosuppression in a murine model of ovarian cancer was significantly reduced upon CTLA-4 blockade, which was largely attributed to a reduction in arginase I activity by these cells [71]. Taken together, these observations suggest that immunoregulatory myeloid cells confer their inhibitory activity via CTLA-4.

##### DCs

Activated conventional human DCs were also found to express CTLA-4 on their surface [72]. Engagement of CTLA-4 on the surface of mature CD1a^+^ DCs may induce a tolerogenic DC phenotype characterized by IDO expression and IL-10 secretion [73]. In line, CTLA-4 engagement was reported to inhibit DC maturation and antigen presentation [74], mediated via activation of signal transducer and activator of transcription (STAT)-3 [75]. Further, a fraction of CTLA-4 was found to be released by DCs via exosomes, and upon binding CD80/CD86 on adjacent APCs, conferred downregulation of these costimulators [76]. Here, Halpert and coworkers demonstrated that silencer RNA-mediated knock-down of CTLA-4 on mature DCs stimulated the proliferation of CD8^+^ T cells and improved melanoma control [76], and that the secretion of micro vesicular CTLA-4 by DCs also regulated Th1 polarization and immunity [76,77] This is in accordance with previous results by Chen and coworkers, who showed that binding of tumor-cell-expressed CTLA-4 with CD80/86 on DCs suppressed their ability to synthesize Th1-polarizing cytokines, thus impairing the differentiation of naïve T cells into IFN-γ secreting Th1 effector cells [78]. Moreover, Tregs, which display constitutive expression of CTLA-4, induced expression of IDO in plasmacytoid (p)DCs upon engagement of CD80/CD86 by reverse signaling [79]. Further, the upregulation of IDO was a consequence of autocrine CTLA-4-induced interferon (IFN)-γ production [80]. It has been demonstrated that IDO inhibits T-cell activation and the activity of T effector cells, and promotes Treg induction and activation by depletion of tryptophan from the extracellular space and by the release of tryptophan degradation products, termed kynurenins [81]. In addition, kynurenins were reported to inhibit the function of IDO-deficient DCs [82]. Altogether, CTLA-4, by binding CD80/CD86, exerts inhibitory effects on DCs, e.g., via induction of iDO, and accordingly, CTLA-4^+^ DCs contribute to spread tolerance.

##### B Cells

Besides DCs, B cells were also reported to express CTLA-4 on their surfaces [83,84]. Direct cell–cell contact with activated T cells [83], or simultaneous stimulation with IL-4 in combination with agonistic CD40-specific antibody or LPS, was sufficient to induce CTLA-4 expression on B cells [85]. Cross-linking of B-cell-bound CTLA-4 in vitro in a T-cell-free environment restrained isotype switching and attenuated antibody production, particularly including IgG, as well as the production of proinflammatory tumor necrosis factor (TNF)-α and IL-8. The role of CTLA-4 in limiting B-cell responses is further exemplified by the hyperactivated B-cell phenotype observed in CTLA-4^−/−^ mice [86]. In light of the physiological role of CTLA-4 as a negative regulator of DC and B-cell activity, further studies are necessary to clarify the role of CTLA-4 on B cells in the context of tumor therapy using ICI.

##### NK Cells

CTLA-4 expression has also been reported for NK-cells upon IL-2 priming, although CTLA-4 expression was largely confined to the intracellular compartment [87]. However, murine NK cells expressed CTLA-4 and CD28 when treated with IL-2 and IL-12. The binding of CTLA-4 has been found to exert inhibitory roles on mouse NK cell function. Using NK cells derived from mice deficient in either CTLA-4 or CD28, it was demonstrated that Fc-CD80 and coculture with stimulated DCs promoted IFN-γ production of NK cells that expressed CD28 only. In humans, it has been observed that NK cells derived from healthy donors, from CTLA-4 haploinsufficient patients, expressed CTLA-4 in response to proinflammatory cytokines [88]. Further, CTLA-4 haploinsufficient NK cells were characterized by strongly diminished cytotoxic activity and IFN-γ production in response to stimulation, as compared to NK cells of healthy donors. These findings indicate that CTLA-4 on NK cells serves to limit NK activity.

##### Tumors

Next to immune cells, CTLA-4 may also be apparent on the surface of melanocytes, and has been detected in many cases of IFNγ-stimulated melanoma cells [89], as well as other types of tumors [90,91]. CTLA-4-expressing breast cancer cells were shown to impair the T-cell stimulatory activity of cocultured DCs by activation of extracellular signal-regulated kinase/STAT3 in the latter [78]. In the same publication, the addition of a CTLA-4-specific antibody to CTLA-4-positive tumor cells directly reduced cell viability. Additionally, it has been shown that incubation of CTLA-4-expressing human tumor cell lines with solubilized CD80 and CD86 induced tumor-cell apoptosis, prompting the authors to suggest a specific targeting of CTLA-4 on tumor cells for therapy [92]. However, this finding has not been confirmed by other studies so far. Taken together, these observations suggest that CTLA-4 on tumor cells promotes immune evasion by inhibiting APCs.

### 4.2. Immunomodulatory Effects of Anti-PD-1/Anti-PD-L1 on Leukocytes and Tumor Cells

The surface receptor PD-1 is mainly expressed on T cells, and its engagement inhibits T cell activation [93]. PD-1 is engaged by PD-L1 and PD-L2, which are expressed by various APCs, such as monocytes/macrophages and DCs, but also by non-lymphoid tissues and particularly by tumors [94]. Binding of these ligands to the PD-1 receptor on T cells causes intracellular signaling in the T cell, which is mediated by the immune-receptor tyrosine-based inhibitory motif of the PD-1 receptor tail. This results in attenuated cytokine production, decelerated cell cycle progression and impaired expression of anti-apoptotic factors, such as Bcl-like protein 1, and of transcription factors associated with effector functions [95].

Monoclonal antibodies that block the PD-1 receptor with high affinity prevent binding of PD-L1/2 and are thereby able to prevent inhibitory T-cell signaling [3]. Thus, blockade of the PD-1/PD-L1/2 axis promotes anti-tumor T cell responses. Currently, six antibodies that block the PD-L1/PD-1 axis have been approved by the FDA for treatment of metastatic melanoma and other types of tumors, due to the induction of durable and persistent clinical responses in various advanced cancer entities (Table 1) [96].

Blockade of PD-1 resulted in even stronger anti-melanoma responses and better patient outcomes while presenting with a better toxicity profile, as compared to CTLA-4 [97]. PD-1-blocking antibodies (nivolumab, pembrolizumab) have been approved firstly for treatment of metastatic melanoma, followed by other solid tumor types, such as lung cancer, renal cell carcinoma and head and neck squamous cell carcinoma [3]. Patients suffering from metastatic melanoma showed a significant prolongation of PFS (median PFS: 11.5 vs. 6.9 vs. 2.9 months when receiving the combination of ipilimumab plus nivolumab, nivolumab alone and ipilimumab alone) and OS (not reached vs. 36.9 vs. 19.9 months) [98,99,100]. Moreover, it has been observed that combined ICI therapy (ipilimumab plus nivolumab) yielded the strongest and most durable anti-tumor response, with five-year PFS rates reaching 36% (vs. 29% for nivolumab and 9% for ipilimumab monotherapy, respectively) and the longest response durability after treatment cessation [100]. However, combined ICI therapy has also shown the highest rate of grade 3 or 4 treatment-related irAEs (59%), whereas nivolumab alone yielded a somewhat better toxicity profile as compared to ipilimumab monotherapy (23% vs. 28%) [100].

Similarly to nivolumab, clinical trials investigating the efficacy of pembrolizumab have confirmed durable anti-tumor activity and tolerability in advanced melanoma, with an estimated median OS of 38.6 months, a median PFS of 16.9 months and 5-year PFS rates of 29% in treatment-naïve melanoma patients [6,101]. More recently, another PD-1-specific antibody (cemiplimab) has been approved for the treatment of advanced cutaneous squamous cell carcinoma, based on the data of clinical trials that have demonstrated robust and durable anti-tumor responses (objective response rate [ORR]: 44%, and observed duration of response >6 months: 68%), while showing an acceptable safety profile [102,103].

**Table 1 cancers-14-01710-t001:** FDA-approved treatment regimen for blockade of the PD-1/PD-L1 axis in tumor therapy.

FDA Approval	PD-1 Antibody	PD-L1 Antibody	References
Advanced and metastatic melanoma	Nivolumab Pembrolizumab		[99] [101]
Metastatic Merkel cell carcinoma		Avelumab	[104]
Advanced cutaneous squamous cell carcinoma	Cemiplimab		[102]
Locally advanced or metastatic urothelial carcinoma	Nivolumab Pembrolizumab	Atezolizumab Durvalumab	[105,106,107,108,109,110]
Metastatic non-small-cell lung cancer (NSCLC)	Nivolumab Pembrolizumab	Atezolizumab Durvalumab	[43,111,112,113,114,115,116]
Metastatic small cell lung cancer	Nivolumab	Atezolizumab	[43,116]
Metastatic pleural mesothelioma	Nivolumab		[117]
Advanced or metastatic renal cell carcinoma	Nivolumab		[118]
Unresectable, locally advanced or metastatic triple-negative breast cancer		Atezolizumab	[44]
Hodgkin lymphoma that has relapsed or progressed after autologous hematopoietic stem cell transplantation	Nivolumab Pembrolizumab		[119] [120,121]
Metastatic or recurrent head and neck squamous cell carcinoma	Nivolumab Pembrolizumab		[122] [123]
Locally advanced or metastatic human epidermal growth factor receptor 2^+^ gastric cancer	Pembrolizumab		[124]
Hepatocellular carcinoma (HCC)	Nivolumab		[125]
Microsatellite instability-high or mismatch repair deficient metastatic CRC	Nivolumab		[45]
	Pembrolizumab		[126,127]
Locally advanced or metastatic esophageal cancer that relapsed on prior therapy	Nivolumab Pembrolizumab		[128] [129]

PD-L1 inhibitors, which block PD-L1 on tumor cells and APCs, have also shown great success in the treatment of advanced solid tumors [130]. There are currently two FDA-approved anti-PD-L1 antibodies (avelumab, atezolizumab) that provide a substantial prolongation of survival and high objective response rates in different advanced cancer diseases. Notably, PD-L1-blocking antibodies produced relatively low rates of grade 3 and 4 treatment-related irAEs, which allows for a specific and safe anti-tumor therapy [96]. In particular, avelumab has demonstrated durable responses and long-term survival outcomes in patients with metastatic Merkel cell carcinoma with an ORR of 33%, a median duration of response of 40.5 months, and a median OS of 12.6 months. Meanwhile, treatment with avelumab has proven to be safe, with only 11% of patients showing grade 3 or 4 treatment-related irAEs [104,131]. Further, it has been observed that the PD-L1-specific antibody atezolizumab significantly improved PFS and OS in patients with advanced PD-L1-positive non-small-cell lung cancer (NSCLC) [114,132]. The PD-L1-blocking antibody durvalumab, which was approved for the treatment of metastatic urothelial cancer in 2017, has been withdrawn recently, since the confirmatory phase 3 trial missed the primary endpoints [110]. Notably, treatment efficacy with PD-1- and PD-L1-blocking antibodies correlated with the extent of PD-L1-expression in the tumor [133], although many patients are refractory towards anti-PD-L1 therapy despite PD-L1 expression by the tumor [96]. On the other hand, it has been found that patients with PD-L1-negative tumors may show a response to PD-1-based immunotherapy, as has previously been reported by Naumann and coworkers for metastatic cervical cancer [134]. Even durable responses may be observed in PD-L1-negative patients, as demonstrated by Kefford and coworkers in the treatment of malignant melanoma with pembrolizumab [135]. However, the ORR of PD-L1-negative patients was substantially lower (6%) as compared to PD-L1-positive patients (51% in melanoma, 36% in third-line small-cell lung cancer [136] and 21% in head and neck cancer [137]). Although the basis for PD-1-inhibitor activity in PD-L1-negative patients is incompletely understood, culminating evidence from multiple clinical trials in different cancer entities indicates that this phenomenon might have a wider biological basis [138]. In particular, it has been proposed that the dynamic expression of PD-L1 both on a temporal and spatial level might account for the observation that PD-L1-negative tumors at the time of histopathological examination showed response to subsequent PD-1-blocking therapy. Hence, dynamic PD-L1 expression limits the reliability of baseline PD-L1 testing [139]. The phenomenon of dynamic temporal PD-L1 expression has been exemplified by two studies that examined PD-L1 expression before and after cancer treatment. These studies showed an increased PD-L1 tumor positivity following chemotherapy in lung cancer patients [140], and following Nivolumab treatment in melanoma patients [141]. The mechanisms of PD-L1 upregulation have been attributed to altered oncogenic signaling, hypoxia signaling, genetic alterations and altered inflammatory signaling [139]. However, comprehensive analysis is lacking so far. The lack of clinical association between levels of PD-L1 expression in the tumor and response to PD-1-targeting checkpoint inhibitors may also be inferred from the potential role of PD-L1 on non-tumor cells, such as macrophages, monocytes, PMNs, DCs and T helper cells. In particular, Liu and coworkers found an association between clinical benefit to PD-1-blocking therapies and PD-L1 expression on macrophages in NSCLC [142]. Therefore, translational studies may not only focus on PD-L1 expression on tumor cells to predict responsiveness towards PD-1-targeting immunotherapy but may consider increasing the predictive strength by assessing PD-L1 expression on specific immune cell populations, such as macrophages, in addition. However, it should be considered that previous reports suggested that immunohistochemical assays per se may lack sensitivity, due to the use of different antibody clones, staining platforms and scoring systems in each institute, which may explain in part the lack of correlation between PD-L1 expression in the tumor and the clinical benefit of PD-1-targeting immunotherapy [143].

#### 4.2.1. Effects of PD-1/PD-L1 Axis Blockade on T Cells and beyond

Owing to the detrimental role for anti-tumor immunity, most studies investigating the mechanisms underlying PD-1/PD-L1 blockade in cancer initially focused on the beneficial effects for CD8^+^ T cell cytotoxicity, due to disruption of inhibitory signaling on the T-cell side, and thereby increased anti-tumor function [144]. However, recent evidence suggests that the outcome of PD-1/PD-L1-blocking ICI therapy may also be conferred in large part by non-CD8^+^ T cells, comprising innate and adaptive immune cell types [145]. In particular, it has been demonstrated that the expression of PD-1 is not only confined to (CD8^+^) T cells, but also shows a significant expression on myeloid cells, B cells and NK cells, as well as on a variety of tumor cells [146]. Moreover, it has been found that PD-L1/2 expression, which has initially been reported as largely restricted to tumor cells and tolerogenic APCs, is also expressed on myeloid cells, B cells and non-immune cells, such as endothelial cells or fibroblasts [147] (Figure 3).

##### T Helper Cells

While PD-1 expression has not been described on naïve CD4^+^ T helper cells, it has been reported on Treg [148] and T follicular helper cells (Tfh) [149] and is particularly observed in Th1 and Th2 cells upon activation in an inflammatory cytokine environment [145,150]. The important functional role of PD-1 for influencing CD4^+^ T-cell dependent immune responses has been demonstrated by Nagasaki and coworkers, who reported on the significance of a PD-1-mediated CD4^+^ T cell anti-tumor immunity in a mouse model of Hodgkin’s lymphoma, which has been abrogated in CD4^+^ T-cell deficient mice [150]. The relevance of T-helper-cell-driven anti-tumor immunity has also been confirmed in humans, since the number of CD4^+^ PD-1^+^ T cells was shown to be strongly correlated with disease progression in NSCLC [151] and chronic lymphocytic leukemia patients [152]. Further, Koretkaas and coworkers reported strong expression of PD-1 on exhausted CD4^+^CD39^+^ T cells, which upon application of PD-1-blocking antibodies showed an enhanced cytokine production and elevated levels of CD40L [153], and induced CD8^+^ T cell proliferation [154].

For several cancer entities, an association of PD-1^+^ Treg with poor survival has been reported, including NSCLC and melanoma [148]. This association has been attributed to the interaction of Treg-bound PD-1 with PD-L1 on CD8^+^ T cells, which suppressed the anti-tumor activity of the latter [155]. PD-1 blockade yielded lower Treg numbers within the TME, accompanied by elevated activity of T effector cells [156]. Additionally, it has been found that in a murine lung cancer model, resistance towards PD-1 blockade was associated with high numbers of Th17 cells, and could be overcome by cotreatment with IL-17 neutralizing antibodies [157]. Of note, T-bet was found to inhibit the expression of PD-1 in Th1 cells, as has been observed in patients with Crohn’s disease [158].

Last, it has been shown in CRC that triggering of PD-1 restricts the differentiation and anti-tumor effector functions of Tfh and results in a diminished production of IL-21, whereas PD-1 blockade increases the effector function of Tfh, thereby enhancing CD8^+^ T-cell responses [159]. Here, Zappasodi and coworkers demonstrated that anti-PD-1 therapy augmented the function of Tfh in a B16 melanoma model [151].

Overall, these reports highlight the outcome of PD-1 blockade on various T-cell populations, and the fact that patient stratification for anti-PD-1-based therapies should also consider CD4^+^ T helper cell subsets. Despite the overall inhibitory activity of the PD-L1/2-PD-1 axis on T helper cells, it was also noted in several disease models that CTL expressed PD-L1 [160]. In turn, PD-L1 was reported as necessary for the expression of anti-apoptotic factors, including B-cell lymphoma-extra-large (Bxl-xl) and CD127. Likewise, in a murine melanoma model, PD-L1 blockade was reported to result in p38-MAPK-dependent induction of apoptosis of CD4^+^ T cells [161].

##### B Cells

Immune-cell expression of PD-1 has been reported first for human B cells [162]. Initial studies demonstrated that PD-1 limited B cell proliferation and antibody production, since PD-1-deficient mice showed increased levels of IgG and IgA and higher numbers of B-cells in the spleen and in the peritoneal cavity [163]. In accordance, binding of PD-1 by PD-L1 or PD-L2 was found to decrease B-cell effector functions, such as the secretion of inflammatory cytokines [162]. Blockade of this interaction restored B-cell effector functions. Hence, the activation state of B cells has been discussed as a prognostic marker in anti-PD-1 treatment [164], but studies investigating the effects of clinically applied antibodies for B cells have not been performed yet. PD-1 has also been shown to regulate germinal center B-cell survival in mice [165]. Additionally, PD-1 signaling determined the survival of plasma cells in mice. Notably, more recently a subset of immunoregulatory PD-1^+^ IL-10-producing B cells has been demonstrated to inhibit T-cell responses via PD-L1 engagement [166]. Further, B cells have also been reported to express PD-L1/2. Of note, it has been shown that B cells attenuate Tfh effector function via the PD-(L)1-signaling axis [167]. Altogether, PD-1-expressing B cells may be inhibited in their activity upon binding of PD-L1/2, which is expressed by regulatory immune cells. Furthermore, B cells may spread tolerance when expressing PD-1 ligands.

##### Monocytes

PD-L1 expression has also been reported for CD14^+^ monocytes, and was found to be positively regulated by TNF-α and IL-6 in systemic lupus erythematosus [168], whereas the application of TGF-ß resulted in a downregulation [169]. It has been shown that PD-L1/2^+^ monocytes produced IL-10 and showed little anti-tumor activity in an in vitro hepatocellular carcinoma (HCC) model [170]. Also, co-cultivation of PD-L1^+^ monocytes with CTL significantly suppressed their tumoricidal activity, whereas PD-1 blockade restored the anti-tumor activity of CD8^+^ T cells. In addition, PD-1 blockade resulted in the expansion of Ly6C^+^ monocytes and DCs within melanoma [171]. In tumor patients, high PD-L1 expression on monocytes was found to correlate with low responsiveness towards PD-(L)1 blockade, and consequently low overall survival in lung cancer [172] and hepatocellular carcinoma [170]. These observations indicate that PD-L1-expressing monocytes present themselves with immunoregulatory functions and inhibit T cells via the PD-L1/PD-1 axis.

##### Macrophages

Macrophages are cells of prognostic importance in many tumor patients [173]. Macrophages can switch between different functional states, currently classified as M1- (T cell activation and APC-like state) and M2- (IL-10 producing, wound healing state) phenotype [174]. Macrophages can also be apparent as TAMs. TAMs are predominantly of an M2-like state, acquired under the influence of the cytokine profile of the TME. TAMs contribute to the immunosuppressive capabilities of the TME, resulting in the association with poor prognosis in cancer [175]. In contrast to initial reports, which found no expression of PD-1 on peritoneal macrophages [176], another study demonstrated a strong expression of PD-1 on macrophages [177]. PD-1 expression was mainly induced following stimulation with toll-like receptor (TLR) ligands and nuclear factor kappa B (NF-κB) activation [178]. In accordance, PD-1 expression in myeloid progenitors was observed at an early stage in colon cancer tumorigenesis, and has been associated with tumor-associated inflammation [171]. Similarly, PD-1 expression was enhanced by macrophages in a murine sepsis model [179].

Since macrophage function is critically regulated by activated T cells via secretion of IFN-γ [180], the role of PD-1 for macrophages is closely linked to the functional state of T cells [181]. Notably, triggering PD-1 on macrophages did affect their tumor infiltration and effector functions [182]. In particular, Diskin and coworkers reported that upon binding of PD-L1^+^ T cells to PD-1^+^ macrophages, an alternative M2-like phenotype was induced in the latter, thus abrogating anti-tumor immunity [183]. In accordance, it has been found that in a murine sarcoma model, PD-1 blockade resulted in a reduction in CD206^+^ M2 macrophages and an increase in inducible nitric oxide- (iNOS) expressing M1 macrophages, characterized by an enhanced anti-tumor potential as evidenced by stronger NF-κB activity and IFN-γ-production [184]. Notably, neutralization of IFN-γ significantly diminished macrophage polarization towards an M1 phenotype, highlighting the importance of IFN-γ-secreting intratumoral T cells, apparently as an indirect effect of PD-1-blocking therapies [185].

In line with these reports, Chen and coworkers observed that triggering of macrophage-bound PD-1 skewed polarization towards an M2 phenotype, whereas the deletion of PD-1 promoted polarization towards an M1 phenotype, which correlated with an enhanced inflammatory cytokine production, intracellular killing ability and phagocytosis [179]. In accordance, Gordon and coworkers showed in a murine colon cancer model that PD-1 blockade enhanced the phagocytotic activity of TAMs and reduced tumor burden [182]. Remarkably, tumor growth was prevented in a murine myeloid-specific PD-1 knockout model of colon cancer, thus demonstrating that PD-1 expression on macrophages may be necessary to drive tumor progression [171]. Moreover, it has been found that myeloid-specific PD-1 expression alters myelopoiesis, thus improving the anti-tumor function of myeloid cells. Lastly, interaction of macrophage PD-1 with PD-L1 decreased IL-12 expression in vitro and in vivo [179]. In vitro, this inhibitory effect was counteracted by anti-PD-1 antibodies.

The clinical relevance of PD-1 expressed by macrophages has further been demonstrated in various human cancer entities, since tumor-infiltrating macrophages showed a significantly higher PD-1 expression compared to macrophages of healthy controls, and the numbers of PD-1^+^ macrophages were associated with disease progression and poor prognosis [186,187]. This finding has been attributed to the immunosuppressive effects conferred by PD-1^+^ macrophages. In accordance, it has been observed that therapeutic blockade of PD-1 promoted tumor infiltration by M1 macrophages, known to exert anti-tumor activity [188]. Furthermore, these studies suggest that PD-1 on macrophages is a critical checkpoint protein that promotes tumor growth. Hence, PD-1-directed therapies may directly act on PD-1^+^ TAMs and macrophage-mediated immunity in cancer. Therefore, anti-PD-1 blockade may not only enhance anti-tumor activity of activated T cells, but may also improve myeloid-cell-driven anti-tumor immunity via enhanced polarization towards a macrophage M1 phenotype and an accumulation of iNOS^+^ TAMs within the TME [189].

In accordance with the association found between a tumor-induced systemic inflammation and the upregulation of PD-1 on macrophages, it has also been reported in various murine cancer models, including B16 melanoma and 4T1 mammary carcinoma, that anti-PD-1/PD-L1 combined with anti-IL-6 [190] or anti-IL-1ß [191] antibodies yielded synergistic anti-tumor effects, thus indicating that co-targeting of the inflammatory TME may amplify the efficacy of anti-PD-1 therapy.

Next to the role of PD-1 expressed by macrophages, the role of PD-L1 on this cell type has also gained increasing interest in recent years. To date, the main bulk of research investigating the role of PD-L1 for macrophages has focused on their disrupted interaction with T cells: Binding of macrophage-bound PD-L1 to PD-1 on T cells resulted in a diminished T-cell proliferation and effector T cell functions, which could be reversed by the application of anti-PD-L1 antibody [192]. Moreover, it has been demonstrated that the blockade of macrophage-bound PD-L1 favored the repolarization of TAMs towards an inflammatory phenotype characterized by enhanced iNOS and CD40 expression [193].

Taken together, engagement of PD-1 on macrophages may promote their conversion towards M2 and TAM, whereas PD-L1-expressing macrophages inhibit T effector cells via PD-1.

##### Polymorphonuclear Neutrophils

In contrast to macrophages and DCs, the role of the PD-1/PD-L1 axis for tumor-infiltrating PMNs is largely unknown. Tumor-infiltrating PMNs were shown to comprise two distinct subpopulations, termed N1 and N2 tumor-associated neutrophils (TANs), exerting either pro- or anti-tumor properties [194]. The expression of PD-L1 on PMNs has been reported to be associated with a pro-tumorigenic phenotype, since PD-L1^+^ PMNs suppressed CD8^+^ T cells, thereby worsening HCC patient survival [195]. In addition, Granuk and coworkers reported for a mouse breast cancer model that binding of PD-L1 on PMNs to PD-1, as expressed by tumor cells, suppressed neutrophil cytotoxicity. In agreement, in murine mammary carcinoma models, blockade of PD-1/PD-L1 interaction enhanced tumor-cell susceptibility to PMN cytotoxicity, thus limiting primary tumor growth and metastatic progression [196]. Lastly, it has been suggested that PMNs may also modulate NK cell immunity via the PD-1/PD-L1 axis [197]. In particular, in colon-carcinoma-bearing mice, binding of PD-L1 on PMNs to PD-1 on NK cells suppressed the anti-tumor immunity of the latter [197]. Altogether, PD-L1^+^ PMNs may inhibit T-cell responses via PD-1 engagement.

##### MDSCs

The expression of both PD-1 and PD-L1 on MDSCs has been reported during infection [198] and in a mouse model of malignant melanoma [199]. As observed for other myeloid cell types, expression of PD-1 and PD-L1 proteins is induced during inflammation. Further, it has been reported by Liu and coworkers that PD-L1 expression is significantly regulated via the pSTAT1-Interferon regulatory factor 1 axis, and that PD-L1 expression is enhanced by the addition of IFN-γ [200].

Interestingly, in colon-carcinoma-burdened mice, a myeloid-specific PD-1 knock-down prevented tumor-induced emergency myelopoiesis, which normally results in the generation of PD-1^+^ MDSCs [171]. Instead, higher numbers of myeloid cell types exerting anti-tumor activity were generated. Hence, it has been suggested that PD-1^+^ myeloid progenitors direct emergency myelopoiesis in favor of the granulocyte lineage. This results in larger numbers of granulocytic MDSCs within the tumor, which may limit the efficacy of checkpoint-inhibitor therapy as has been observed in various cancer settings. In particular, it has been shown that the frequency of tumor-infiltrating MDSCs increased in tumor biopsy lesions of patients with metastatic melanoma, which have been treated with anti-PD-1 therapy [201]. PD-1/PD-L1 blockade may therefore indirectly drive checkpoint resistance by expanding and recruiting MDSCs. In accordance, it has been shown that the frequency of granulocytic MDSCs correlates with poor response and survival in various cancer entities [202]. However, reports from patients with head and neck squamous cell carcinoma did not confirm these observations, and rather showed a reduced infiltration of granulocytic MDSCs post-anti-PD-1 therapy [203]. Further, it has been found that treatment with CTLA-4-blocking antibodies also reduced the frequency of granulocytic MDSCs three weeks after the initial ipilimumab dose, and was followed by a reduction in Arginase-1-expressing MDSCs [204].

PD-L1 blockade has, however, been shown to interfere with the immunosuppressive capacities of MDSCs. Specifically, Noman and coworkers observed an increased T-cell proliferation and IFN-γ production upon treatment with anti-PD-L1 antibodies [205,206]. Further, a high expression of PD-1/PD-L1 on MDSCs has been reported to correlate with high rates of proliferation, resulting in the expansion of MDSCs within the TME [199]. Overall, it has been found that PD-1 substantially impacted the differentiation, the regulation of the function and the proliferation of MDSCs.

##### DCs

As mentioned above, expression of PD-L1/2 has initially been reported as largely confined to tumor cells and tolerogenic DCs [207], whereas PD-1 has been considered as primarily expressed by T cells [208]. However, both types of receptors can be expressed by additional immune cell types [147]. Zhao and coworkers demonstrated that PD-L1 on APCs formed heterodimers with CD80, which did not impair the agonistic activity of CD80 on CD28 in the course of T cell activation, but protected CD80 from trans-endocytosis by CTLA-4 [209]. By contrast, Butte and coworkers reported that binding of PD-L1 to CD80 restricted the costimulatory signal for T cell-bound CD28, thus inhibiting T-cell responses [210]. Hence, further research is needed to better understand the role of PD-L1/CD80 engagement on APCs. Interestingly, the administration of atezolizumab, a PD-L1 inhibitor, resulted in an attenuated surface expression of CD80, which has been attributed to an impaired formation of PD-L1/CD80 heterodimers, prone to CTLA-4-mediated trans-endocytosis of CD80. In agreement, concomitant application of ipilimumab restored CD80 levels. PD-L1-mediated protection of CD80 from trans-endocytosis by CTLA-4 could also account for the observation that K562-cell-based artificial APCs yielded a stronger activation of CD8^+^ T cells when these APCs were engineered to co-express CD80/CD86 and PD-L1, as compared to CD80/CD86 expression only [211].

As demonstrated in a mouse model of ovarian cancer, tumor-infiltrating DCs de novo expressed PD-1 [212], which was mimicked in vitro by applying the anti-inflammatory cytokine IL-10 to bone marrow-derived DCs [213]. Of note, antibody-mediated blockade of PD-1 on DCs resulted in elevated production of IL-10 both in vitro and in a model of ovarian cancer [213]. Accordingly, only the combined application of anti-PD-1 and anti-IL-10 antibodies was effective in delaying tumor growth.

Administration of IL-10 in the course of GM-CSF/IL-4-induced generation of human DCs from monocytes was shown to yield a cell population reminiscent of primary MDSC [214]. Among other markers, these IL-10-induced-MDSCs and tumor-infiltrating DCs were characterized by the expression of both PD-1 and PD-L1. Both cell types inhibited T-cell activation when titrated to DC/T cell cocultures. However, pretreatment of either cell population with PD-1- or PD-L1-blocking antibodies was sufficient to counteract their suppressive function. In accordance, PD-1 was reported to impair the survival of DCs [215] and the generation of IL-12 and TNF-α [216], thus indirectly inhibiting CTL activity. Further, it has been shown that PD-1 signaling in DCs also inhibited NF-κB signaling and MHC-I expression [217].

Garris and coworkers showed that tumor-infiltrating DCs that generated IL-12 were necessary for anti-PD-1 induced tumor regression, which was not observed after DC depletion in a mouse fibrosarcoma model [218]. The immunosuppressive role of PD-1 on DCs was confirmed by Lim and coworkers by showing that the intratumoral transfer of PD-1-deficient DCs led to an enhanced priming of CD8^+^ T cells, and in consequence reduced tumor growth in an in vivo model of HCC [219].

DCs, besides other types of APC, were also shown to express PD-L1, particularly in the setting of chronic inflammatory conditions, as present in the TME [220]. Here, expression of PD-L1 by DCs has been found to directly limit T cell responses, thus controlling immune homeostasis [221]. In the context of tumor progression, in a murine colon tumor model, Peng and coworkers showed that a DC-specific PD-L1 deletion restricted tumor growth in mice as effectively as systemic PD-L1 knock-out, suggesting that DCs and DC-bound PD-L1 are critical in regulating T-cell responses upon PD-L1 immunotherapy [222]. In accordance, Oh and coworkers demonstrated in a murine colon tumor model that a DC-specific deletion of PD-L1, in particular, resulted in an enhanced infiltration by CD8^+^ T cells, whereas proliferation was not enhanced [207]. DC expression of PD-L1 was further found to induce Treg numbers and augment Treg immunosuppressive activity in a human model of xenogeneic graft-versus-host disease [223]. Therefore, it has been suggested that DC-expressed PD-L1 may show a stronger correlation with clinical response to anti-PD-1 therapy as compared to tumor-expressed PD-L1, based on results obtained for various human tumor types [224].

Taken together, both the finding that PD-L1 may also yield anti-apoptotic effects in activated T cells and its complex contribution to the costimulatory activity of APCs, such as DCs, suggest that its blockade may yield unexpected consequences that must be taken into account, especially with regard to the ongoing debate on the reasons for the resistance of a considerable number of patients towards ICI treatment [225].

##### Other Granulocytes

Eosinophils, basophils [226] and mast cells [227] have been reported to express PD-L1. A considerable number of translational studies have reported on the role of eosinophil abundance to predict response to PD-1-focused checkpoint inhibition. Umansky and coworkers showed that that an early accumulation of eosinophils in the blood and in the tumor tissue of patients with metastatic melanoma treated with ICI correlated with a favorable response [228]. Hence, the accumulation of eosinophils was considered a novel predictive marker to checkpoint-inhibitor therapy [229]. These results have been confirmed not only for metastatic melanoma [230], but also uveal melanoma [231] and other solid cancers [232]. Moreover, eosinophil accumulation has been reported in a B16 melanoma mouse model, and was considered as an early and persistent inflammatory host response [233]. Despite numerous studies reporting on the association of eosinophil accumulation and cancer response, it has not yet been clarified whether eosinophils contribute to tumor immunity or are mere bystander cells. In favor of the former hypothesis, Carretero and coworkers demonstrated several mechanisms of eosinophil-dependent tumor cell killing: First, in a murine melanoma model, tumor-homing eosinophils released chemokines, which in turn attracted T effector cells resulting in tumor eradication [234]. Second, tumor-infiltrating eosinophils mediated M1 polarization of TAMs. This caused a normalization of the tumor vasculature by lowering its leakiness and by elevating vascular cell adhesion protein 1 expression.

As opposed to the positive association found between the relative abundance of eosinophils and the response to PD-1 blockade, Somasundaram and coworkers recently demonstrated in a humanized melanoma model that a strong tumor infiltration by mast cells correlated with resistance to anti-PD-1 therapy [235]. Moreover, tumor-secreted TNF-α enhanced PD-L1 expression by mast-cells. A potential mechanism mediating resistance towards PD-1 blockade has been suggested by Hirnao and coworkers, who observed in a murine contact hypersensitivity model that PD-L1 on mast cells suppressed the activation and function of effector CD8^+^ T effector cells via engagement of PD-1 [236].

In light of the increasing evidence on the role of the PD-1/PD-L1 axis for eosinophils, basophils and mast cells, it would be important to decipher in more detail the role of PD-1 in regulating these cell populations under pathophysiological conditions.

##### NK Cells

NK cells are a subset of innate lymphocytes that exert anti-tumor cytotoxicity [237]. The state of NK cell activity is regulated by the ratio of inhibitory and activating receptors that engage different conserved MHCI motifs [238]. There are conflicting reports on whether NK cells show a significant expression of PD-1 in cancer patients [239]. However, recent evidence indicates that PD-1 expression by NK cells may be upregulated during inflammatory conditions within tumors [240] and that PD-1^+^ NK cells are associated with poor survival of cancer patients [202]. Among others, Vari and coworkers have reported that PD-1 expression impaired NK cell effector functions [241]. In this regard, Trefny and coworkers have shown for NSCLC patients that PD-1^+^ NK cells co-expressed more inhibitory receptors, resulting in an impaired anti-tumor activity, which could be restored by ICI treatment [242]. In agreement, binding of PD-1 to PD-L1 has been found to dampen NK-cell mediated anti-tumor immunity in several murine tumor models [243]. Further, PD-1 engagement was shown to induce apoptosis, whereas PD-1 blockade activated the PI3K pathway in NK cells, thus driving NK cell infiltration and cytotoxicity [244]. Moreover, it has been reported that glucocorticoids increased PD-1 expression on human NK cells, which subsequently acquired an immature, dysfunctional CD56^high^ phenotype [245]. In contrast to these prior observations, Dong and coworkers reported that tumors may also induce PD-L1 expression in NK cells, which in turn results in augmented and long-lasting cytotoxic activity in these cells, as assessed in a murine T lymphoma model [246]. Altogether, engagement of PD-1 on NK cells may affect their viability and cytotoxic activity.

##### Innate Lymphoid Cells

Innate lymphoid cells (ILCs) are innate lymphocytes that are located at barrier surfaces and lack antigen receptors, but generate soluble mediators in response to stimulation [247]. Innate leukocytes are classified into ILC1-3, NK cells and lymphoid tissue inducers, as suggested by Spits and coworkers [248]. A recent study suggests that ILC2 cells express significant levels of PD-1 and may contribute to anti-tumor immunity, as shown by Moral and coworkers in a murine pancreatic tumor model [249]. In this regard, Jacqulot and coworkers demonstrated that combined treatment with IL-33 to activate ILC2 plus anti-PD-1-blocking antibody strongly enhanced the overall melanoma response [250]. Further, it has been found that checkpoint molecules on ILC3, such as PD-1 and TIM-3, may regulate cytokine secretion and immune tolerance [251]. These findings indicate that ILCs may also express PD-1, which upon engagement exerts inhibitory effects. Further studies are required to elucidate in detail the role of the PD-L1/PD-1 axis in ILCs in tumor immunity.

##### Tumor Cells

PD-1 is also expressed on cancer cells, such as in melanoma [252], NSCLC [253] and hepatocellular carcinoma [254]. In the case of the latter study, the expression of high amounts of PD-1 in the tumor resulted in a lower disease-free and OS of patients. In the same study, by using immune-deficient (NOD/SCID) mice, a correlation was shown between in vitro tumor cell viability and PD-1 expression, as well as between in vivo tumor growth and PD-1 expression, which pointed to a PD-1-dependent mechanism independent from adaptive immunity. Despite the absence of a significant contribution of adaptive immune cells in this study, the authors could not exclude indirect immune-mediated effects, which could also explain the association found between tumor growth and PD-1 expression levels. As with HCC, in melanoma, the engagement of tumor cell-bound PD-1 led to increased melanoma growth in immune deficient RAG^−/−^ mice [255]. The proliferation-promoting effect of PD-1 was also visible in the case of PD-1 overexpression, and was reverted by blocking anti-PD-1 antibodies as well as PD-1 specific shRNA. In NSCLC, the effect of PD-1 was inverse: treatment with the anti-PD-1 antibody pembrolizumab resulted in an increased tumor growth rate in comparison to pre-treatment tumor growth [253]. This effect of PD-1 was confirmed using immune-deficient NSG mice. Transplanted tumors of a PD-1-expressing NSCLC cell line showed increased growth and viability after treatment with PD-1 antibodies, whereas overexpression of PD-1 in the same cell line led to decreased viability. Altogether, these results point to a PD-1/PD-L1-dependent mechanism that is independent from adaptive immunity, and may offer an explanation for the differential responses of different tumor entities to anti-PD-1/PD-L1 observed in the clinic.

Expression of the PD-1 ligands PD-L1/2 on tumor cells has been shown in many cancer entities. Here, PD-L1/2 confers inhibition of effector T-cell function and thus dampens anti-tumor immunity [256]. Additionally, it has been shown that PD-L1 confers pro-survival signals in tumor cells [257].

Moreover, emerging evidence supports the possibility of PD-1-independent activities of PD-L1 in malignant cells, such as PD-L1 reverse signaling, which may support malignant cell growth, proliferation and metabolism, as assessed in Hodgkin lymphoma [258], and may also contribute to the clinical efficacy of PD-1 blockade therapy [259]. In this regard, it has been reported that tumor-cell autonomous PD-L1 reverse signaling promotes the epithelial mesenchymal transition (EMT) in human esophageal cancer [260] and induces proliferation and survival in melanoma and ovarian cancer [261]. Further, it has been found that PD-L1 reverse signaling protects B16 melanoma cells from interferon cytotoxicity, thus promoting tumor progression [262]. Notably, PD-L1 reverse signaling has previously also been described on immune cells, such as dendritic cells and macrophages, however, the role of PD-L1 reverse signaling here has been less well characterized to date, and is the subject of ongoing research in the field [263].

To sum up, engagement of PD-1 and PD-L1 on tumor cells may promote their growth, and PD-L1 may serve to impair the anti-tumor activity of infiltrating PD-1-expressing T effector cells.

## 5. Emerging Immune Checkpoints and Their Impact on Non-T-Cell Immunity

As outlined above, only a fraction of tumor patients show sustained responses towards treatment with CTLA4- and PD-1/PD-L1-blocking ICI. In this regard, compensatory upregulation of distinct ICs in response to ICI treatment may contribute to non-responsiveness [264]. Due to the limitations of current ICI treatment options, the suitability of antibody-mediated blockade of additional T-cell-expressed ICs for tumor therapy has been tested in numerous clinical trials [265,266,267]. Here, treatment of metastatic melanoma with the anti-LAG3 antibody Relatlimab in combination with nivolumab has shown promising results in a phase II/III trial, which may result in FDA-approval [268]. However, similarly to CTLA-4 and PD-1, blockade of either IC may yield unwanted immunological side effects, as deduced from their physiological function summarized in Appendix A.

CD96 [269] expression was found to be restricted to T cells [270] and NK cells [271], and is ectopically expressed by acute myeloid leukemia (AML) cells [272]. CD96 binds CD155, which is expressed by innate and adaptive immune cell types. Antibody-mediated blockade and genetic deficiency of CD96 was shown to enhance CTL activities in several mouse tumor models [273]. On the contrary, cross-linking of CD96 was demonstrated to promote CTL activation, and CD96-deficient CTLs were demonstrated to exert impaired anti-tumor activity [274]. Conflicting results were also reported for the functional state of NK cells: whereas CD96/CD155 ligation was demonstrated to enhance the cytotoxic activity of NK cells [271], in CD96-deficient mice used for melanoma studies, NK cells displayed elevated anti-tumor activity due to enhanced IFN-γ production [275].

CD272 (BTLA, B and T lymphocyte attenuator) [276] was detected on T cell populations [277,278,279,280,281], as well as B cells and myeloid cell types [282]. Binding of CD272 to HVEM (herpes virus entry mediator) [283], which is widely expressed by leukocytes [284,285,286,287,288,289] and non-immune cell types like epithelial cells (EC) [287], was demonstrated to confer inhibition of B-cell [290] and T-cell [277,282] activation. Moreover, CD272/HVEM ligation was also reported to inhibit the activity of innate immune cell types [291,292]. However, studies in a graft-versus-host disease model suggested that CD272 also served to enhance T-cell survival [293].

CD276 (B7-H3, B7 homolog 3) belongs to the B7 family of cell-surface receptors [294], and is expressed by activated DCs [295,296] and PMNs [297], but also various non-immune cell types [298,299,300] and tumors [301]. Binding to its receptor TLT-2 (triggering receptor expressed on myeloid cells-like transcript 2) [302,303], apparent on myeloid cells and B cells [304,305] as well as T-cell populations [302], yielded different results, either inhibiting [306,307] or enhancing [295,302,303,308] T-cell activity. Interestingly, in macrophages, TLT-2 expression may be required for IL-6 expression [304] and may enhance phagocytosis of apoptotic cells via engagement of phosphatidyl serine (PS) [309].

CD278 (ICOS, inducible T cell costimulator) [310] is expressed by T-cell populations [311,312] and various innate immune cell types ([313,314,315]. Binding to its cognate ligand CD275 (ICOSL, ICOS ligand), another B 7 family member [316], which is apparent on DCs [317,318,319], B cells [320], several non-immune cells [321,322,323,324] and tumor cells [325], caused Treg expansion [323,326,327,328], IL-10 release by activated T cells [318] and attenuated B-cell activation [329,330,331]. Other studies demonstrated that CD278/CD275 ligation may promote Th2 [332,333], Th17 [315] and Tfh [311] polarization, and may promote overall T cell activation [317,334,335], but may also yield activation of innate immune cell types, including DCs [336,337], natural killer T (NKT) cells [313] and ILCs [314], and inhibit tumor cell migration [338]. In humans, CD28/CTLA-4 were identified as additional ligands of CD278, and engagement enhanced T-cell activity [339].

LAG-3 (lymphocyte activation gene 3) [340] is widely expressed by T cells [341], including Tregs [342], B cells [343] and various innate immune cell types [344,345,346]. LAG-3 was reported to bind MHC-II [347], which is largely restricted to professional APCs [348], the carbohydrate Galectin-3 [349] and hepatocyte-specific FGL1 (fibrinogen-like protein 1) [350,351]. LAG-3/Galectin-3 interaction was repeatedly demonstrated to inhibit T-cell proliferation [341,349,352] and the expansion of plasmacytoid (p)DCs [344,349]. Somewhat contradictorily, LAG-3 deficiency [353] and the binding of LAG-3 to MHC-II [354,355] were reported to promote DC activation. NK cells [356] and NKT cells [346] devoid of LAG-3 showed attenuated activity and proliferation, respectively. In accordance, recent studies demonstrated strong and durable anti-tumor immune responses in patients receiving the anti-LAG3 monoclonal antibody Relatlimab for metastatic melanoma [268].

TIGIT (T cell immunoglobulin and immune receptor tyrosine-based inhibitory motif domain) [357] is expressed by T cell populations [358] and NK cells [359] and binds both CD112 [360], which is expressed by DCs [361] and various non-immune cell types [362,363], and CD155 [358]. By using solubilized TIGIT, engagement of CD155 was demonstrated to result in Treg activation [364] and inhibition of DCs [358] and NK cells [359].

TIM-3 (T-cell immunoglobulin and mucin domain 3) [365] is expressed by T cells [366,367,368], but also DC populations [369,370] and NK cells [371]. Triggering of TIM-3 promoted Treg activation [372] and inhibited DC [369,373], macrophage [374] and NK cell [369] activities. Binding of TIM-3 to Ceacam-1 (carcinoembryonic antigen related cell adhesion molecules 1) [375], which is expressed by T cells, B cells, PMNs [376], non-immune cells [377,378] and tumor cells [379], was shown to confer T effector cell exhaustion [380]. Similarly, engagement of TIM-3 by the carbohydrate Galectin-9 [375], generated by various cell types [381,382,383,384], induced Th1 effector cell apoptosis [375]. On the contrary, binding of TIM-3 to PS [385], characteristic for apoptotic cells [386], enhanced T-cell activity [387]. With regard to other cell types, TIM-3/Galectin-9 ligation was reported to either promote DC [388] and NK cell [389] activity, and to favor expansion of DCs and NKT cells [390]. However, in another report, TIM-3/Galctin-9 interaction was shown to impair NK cell activity [391]. Of note, binding of TIM-3 to the endogenous danger signal high-mobility group protein B1 (HMGB1) [392], released by necrotic cells and activated PMNs [393], impaired DC activity [370].

VISTA (V-set immunoregulatory receptor) [394] is apparent on T cells and different innate immune cell types [395,396,397]. It engages both Galectin-9 [398] at acidic pH values PSGL-1 (P-selectin glycoprotein ligand-1) [399], which is broadly expressed by leukocytes [400,401,402,403,404] and tumor cells [405], and VSGI3 (V-Set and Immunoglobulin domain containing 3) [406], which is restricted to some non-immune cell types [407,408]. Triggering of VISTA exerted broad inhibitory effects on T (effector) cells [395,398,409] and promoted Treg induction [410,411]. Concerning other cell types, activation of both monocytes [412] as well as MDSCs [397] was observed.

## 6. Conclusions/Perspectives

The clinical success of ICI in the past decade has highlighted the central role of the immune system in cancer control. As first observed in metastatic melanoma, treatment with ICI can reinvigorate efficient and durable anti-cancer immunity [5], and up to now has paved the way for this kind of treatment as a standard of care in a number of malignancies [413]. T cells undoubtedly constitute a crucial element in effective anti-tumor immune responses. However, as summarized in this review, the well-established ICs PD-(L)1 and CTLA-4, as well as emerging ICs, also regulate numerous other immune-cell types.

The limited clinical success of ICI therapy may be explained in part both by its detrimental effects on non-intended target cell populations, as discussed in this review, and the compensatory upregulation of non-targeted ICs [414]. Moreover, any ICs and many of the corresponding ligands have been shown to be apparent also in a soluble form in serum, generated either by proteolytic cleavage of alternative gene splicing [415]. Soluble ICs serve as biomarkers and prognostic factors, e.g., in tumor patients, are dynamically regulated, e.g., in response to tumor therapy, and exert biological functions [416]. Therefore, soluble ICs (and their ligands) may also contribute to the limited success of ICI therapy in many patients so far.

Ongoing preclinical and clinical studies focus on the development and evaluation of combination treatment regimens that include the combined application of distinct ICI [265] as well as other biologicals and pharmaceuticals [417], raising the necessity of investigating the potential unwanted side effects. Further, any antibody-mediated therapeutic approach offers the possibility that the constant Fc portion via the engagement of Fc receptors expressed by various (innate) immune cell types may induce antibody-dependent cytotoxicity (ADCC) and classical complement activation [418]. Depending on the Ig subtype, this effect may be negligible [419], as confirmed, for example, for the PD-1-targeting ICI nivolumab and pembrolizumab [420], preventing depletion of PD-1-expressing T cells. However, ADCC may be favorable in the case of antibodies that target ICs predominantly expressed on immunoregulatory cell types and tumor cells, such as PD-L1 [421]. In accordance, the human IgG1 PD-L1-targeting antibody Avelumab was demonstrated to evoke ADCC-mediated killing of breast-cancer cells by NK cells [422]. Therefore, careful design of ICI with regard to Fc-mediated immune effects may have a strong impact on therapeutic efficacy.

Altogether, it is critical for the scientific and clinical communities to integrate their findings on the distinct roles of ICs in different immune cell subsets to understand the potential limitations of ICI therapy, with the aim of successfully using combination therapies to overcome irAEs and unresponsiveness towards treatment.

## Figures and Tables

**Figure 1 cancers-14-01710-f001:**
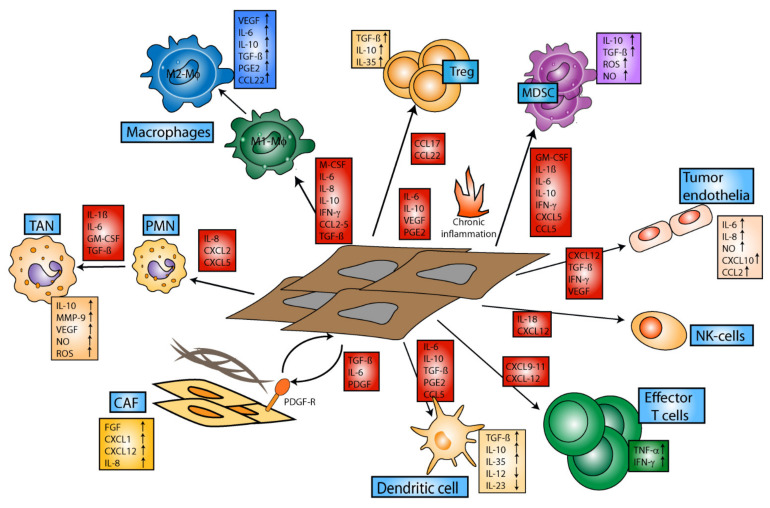
Tumor-induced immune modulation within the tumor microenvironment. Tumor cells secrete various cytokines, chemokines and soluble factors that generate a chronic inflammatory state within the TME that favors the accumulation of immunosuppressive cell types and impairs effector T cell function. Additionally, both immunosuppressive and effector immune cells, as well as non-immune cells such as cancer-associated fibroblasts and endothelial cells, contribute to the maintenance of this inflammatory micromilieu, hampering an effective anti-tumor immune response. Abbreviations: CAF = cancer-associated fibroblasts; IFN-g = interferon-gamma; Mo = macrophages; MDSCs = myeloid-derived suppressor cells; MMP-9 = matrix-metallopeptidase 9; NO = nitric oxide; PDGF-R = platelet-derived growth factor receptor; PGE2 = prostaglandine E2; PMN = polymorphonuclear neutrophils; ROS = reactive oxygen species; TAN = tumor-associated neutrophils; TGF-ß = transforming-growth factor beta; TNF-α = tumor necrosis factor alpha; VEGF = vascular endothelial growth factor.

**Figure 2 cancers-14-01710-f002:**
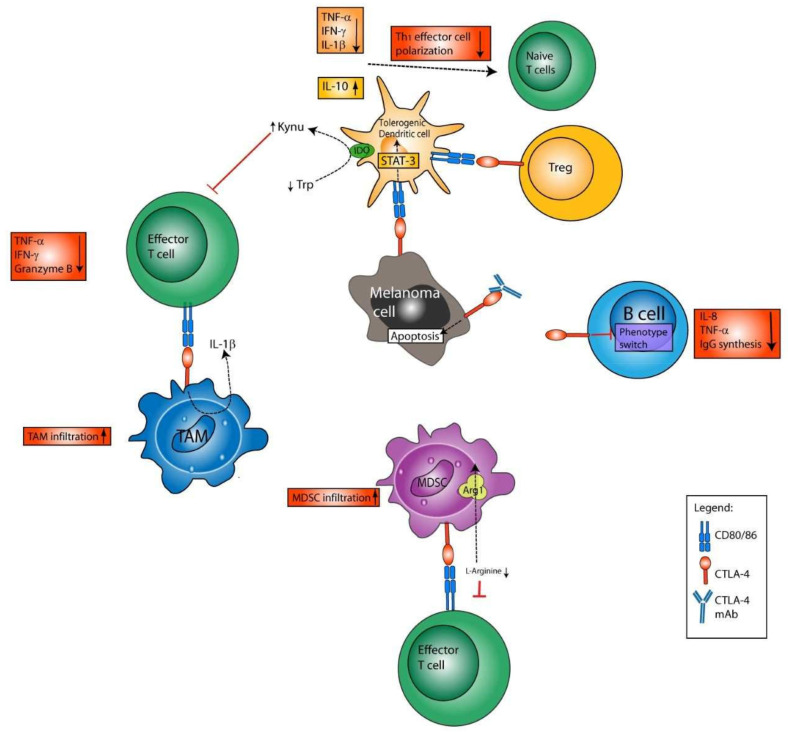
Immunomodulatory role of cytotoxic-T-lymphocyte antigen 4 (CTLA-4). CTLA-4 is expressed by regulatory T cells (Tregs), tumor-associated macrophages (TAM), myeloid-derived suppressor cells (MDSCs), dendritic cells (DCs), B cells and melanoma cells. The interaction of CTLA-4 on melanoma cells, as well as Tregs, with DC-bound CD80/CD86 resulted in the polarization of DCs towards a tolerogenic phenotype. Such DCs conferred depletion of tryptophan (Trp) from the microenvironment via IDO activation, and at the same time enhanced levels of Trp degradation products, termed kynurenins (Kynu). Both Trp depletion and Kynu impair activation/polarization of naïve T cells towards Th1 T effector cells and effector T cell functions. Further, tolerogenic DCs generate soluble anti-inflammatory agents, such as IL-10, which also contribute to T-cell impairment. Further, binding of CTLA-4 on TAMs and MDSCs with CD80/CD86 on T effector cells has been found to impair anti-tumor activity and enhance the infiltration of TAMs and MDSCs into the tumor microenvironment. Lastly, the binding of CTLA-4 on B cells, presumably via CD80/86 on activated T cells (not shown), has been found to reduce antibody synthesis and Ig class switch.

**Figure 3 cancers-14-01710-f003:**
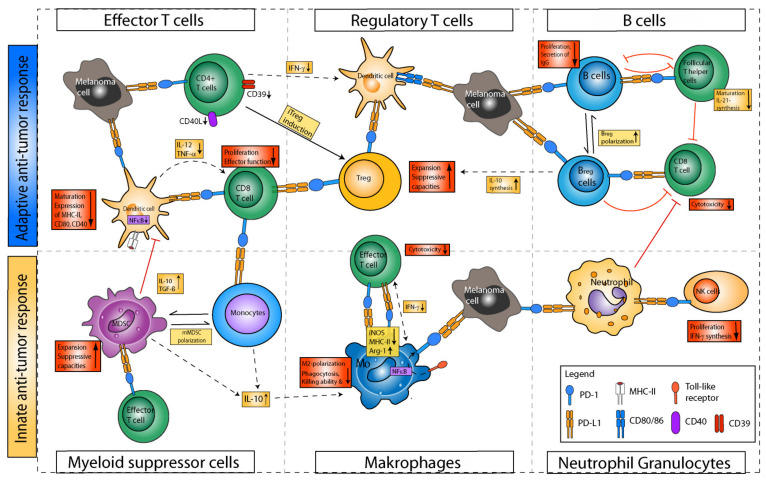
Immunomodulatory effects of the PD-1/PD-L1 axis. Besides CD8^+^ T cells, expression of checkpoint proteins PD-1 and PD-L1 has been reported on CD4^+^ T helper cells, regulatory T cells (Treg), B-cells, innate lymphoid cells, NK cells and various myeloid cells, such as dendritic cells (DC), macrophages, myeloid-derived suppressor-cells (MDSCs), polymorphonuclear neutrophils (PMNs) and monocytes. Overall, PD-1 expression on these immune cell types has been associated with an immunosuppressive phenotype impairing effector T cell functions, both directly, via ligand-–receptor interactions, and indirectly, by creating an immunosuppressive tumor microenvironment. For the majority of studies, these observations were made particularly under the conditions of an inflammatory microenvironment. In this schematic illustration, immunosuppressive activities conferred upon ligation of PD-1/PD-L1 are illustrated in bright red, while cytokine secretions induced via PD-1/PD-L1 signaling are pictured in yellow boxes. Indirect effects of PD-1/PD-L1 ligation are shown in dashed lines, while direct effects are illustrated without dashes.

## Data Availability

Not applicable.

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
