# Peer review of "Immunomodulatory Properties of Immune Checkpoint Inhibitors—More than Boosting T-Cell Responses?"

_cancers, 2022, doi:10.3390/cancers14071710_

Round 1

Reviewer 1 Report

In this review, the authors discuss the role of ICI-based therapies in enhancing non-CD8+ T cell-based immune responses within cancers. They aim to outline the expression and role of CTLA-4 and PD-1/PD-L1 proteins in T cells, other types of immune cells and tumor cells. Besides, the authors integrate the expression pattern and immunological functions of ICs in different immune cell subsets. Although the authors concluded an impressive amount of contents, some parts of the manuscript lack in-depth discussion or conclusions, making it not easy to catch the main points. To make this review more readable and increase its scientific merit, they are encouraged to further strengthen the manuscript by addressing the following questions and comments.

Major points:

  1. In the part of Tumor-induced immune modulation, the authors describe in great detail that cancer cells may secrete cytokines, chemokines or affect other cells in the tumor environment. If possible, draw a figure to summary these contexts for readability.
  2. It has been reported that many immune checkpoint proteins have soluble forms, and these soluble forms of immune checkpoints have also been confirmed to have positive and negative effects on immune response. These points can be added in the manuscript to increase the novelty of this article.
  3. It is recommended to add a conclusion in each paragraph of review article. Such as: chapter 4.1.2.1, chapter 4.1.2.2, chapter 4.1.2.3…please check them thoroughly.
  4. Some descriptions lack the references, please add them in. Such as: Line 134-135, Line 464-467, Line 476-477…
  5. In Line 143-146, the information related to the rate of patients seem to be not found in the reference 34. The similar problem is appeared in reference 80 of Line 310. Please check it.
  6. The content/legend of Figure 1 seems to be not well introduced in the content of the manuscript. For example: 1. In the legend, both Trp and Kynu impaired activation/polarization of naïve T cells, but in the illustration, only Trp plays a suppressing function. 2. CTLA-4 expressed in B2 cells is not reflected in the manuscript.
  7. The illustration is brilliant and colorful in Figure 2, but it would be better if the authors could simplify or rearrange it.
  8. The Supplementary table 1 is a bit messy and suggested to be re-presented in a more concise way.
  9. In Table 1, it is better to add the references for the credibility.
  10. In Line 666, the description is confused. “Innate leukocytes are classified into five groups named ILC1-3, NK cells and lymphoid tissue inducers as suggested by Spits and coworkers [186].” Only three groups are mentioned.

Minor points:

  1. Thoroughly check the spelling mistakes, such as: line88 (“PD-1 ligand (PD-L)1”), line99 (“MDSC are”), line 222 ("…induction ad functional…")…
  2. The reference format is inconsistent in the Supplemental file, please check it. Such as: Lines 104-107 and 197-199.
  3. In line 110, “vascular endothelial growth factor; VEGF” is repeated in line 94.

Author Response

Reviewer 1

In this review, the authors discuss the role of ICI-based therapies in enhancing non-CD8+ T cell-based immune responses within cancers. They aim to outline the expression and role of CTLA-4 and PD-1/PD-L1 proteins in T cells, other types of immune cells and tumor cells. Besides, the authors integrate the expression pattern and immunological functions of ICs in different immune cell subsets. Although the authors concluded an impressive amount of contents, some parts of the manuscript lack in-depth discussion or conclusions, making it not easy to catch the main points. To make this review more readable and increase its scientific merit, they are encouraged to further strengthen the manuscript by addressing the following questions and comments.

Major points:

In the part of Tumor-induced immune modulation, the authors describe in great detail that cancer cells may secrete cytokines, chemokines or affect other cells in the tumor environment. If possible, draw a figure to summary these contexts for readability.

We have added an according schematic overview (Figure 1).

It has been reported that many immune checkpoint proteins have soluble forms, and these soluble forms of immune checkpoints have also been confirmed to have positive and negative effects on immune response. These points can be added in the manuscript to increase the novelty of this article.

We agreee with the reviewer on the overall importance of soluble immune checkpoint proteins. However, due to the large number of studies that focused on the biological relevance of soluble immune checkpoint proteins (and their ligands) we refrained from detailled discussion of this highly interesting issue. Instead, we refer to the importance of this subject in a section of the Conclusions/Perspective section and refer to review articles which intensively discuss this subject.

It is recommended to add a conclusion in each paragraph of review article. Such as: chapter 4.1.2.1, chapter 4.1.2.2, chapter 4.1.2.3…please check them thoroughly.

Conclusions have been added.

Some descriptions lack the references, please add them in. Such as: Line 134-135, Line 464-467, Line 476-477…

References have been added.

In Line 143-146, the information related to the rate of patients seem to be not found in the reference 34. The similar problem is appeared in reference 80 of Line 310. Please check it.

We have adjuste the text accordingly.

The content/legend of Figure 1 seems to be not well introduced in the content of the manuscript. For example: 1. In the legend, both Trp and Kynu impaired activation/polarization of naïve T cells, but in the illustration, only Trp plays a suppressing function. 2. CTLA-4 expressed in B2 cells is not reflected in the manuscript.

The figure and the figure legend have been revised accordingly.

The illustration is brilliant and colorful in Figure 2, but it would be better if the authors could simplify or rearrange it.

This figure (Figure 3) has been altered as suggested.

The Supplementary table 1 is a bit messy and suggested to be re-presented in a more concise way.

Supplemental table 1 has been revised.

In Table 1, it is better to add the references for the credibility.

References have been added in an additional column.

In Line 666, the description is confused. “Innate leukocytes are classified into five groups named ILC1-3, NK cells and lymphoid tissue inducers as suggested by Spits and coworkers [186].” Only three groups are mentioned.

Corrected.

Minor points:

Thoroughly check the spelling mistakes, such as: line88 (“PD-1 ligand (PD-L)1”), line99 (“MDSC are”), line 222 ("…induction ad functional…")…

Spelling mistakes have been corrected.

The reference format is inconsistent in the Supplemental file, please check it. Such as: Lines 104-107 and 197-199.

The reference format has been revised.

In line 110, “vascular endothelial growth factor; VEGF” is repeated in line 94.

Corrected.

We thank the reviewer for his valuable comments and hints.

Reviewer 2 Report

Immune checkpoint inhibitors have marked the treatment of cancer. Their main role is blocking T-cell inhibitory signalling axis. However, the surface molecules aimed at by ICI are expressed by other immune and non-immune cells. The authors provide an overview of these cells, together with the evidence for their importance in response to ICI therapy. The review provides a rich resource to the literature investigating checkpoint inhibitor expression.

In chapter 2 a rather extensive description of melanoma is given. However, in the further text, other cancer types are mentioned. Sometimes it is not clear without going back to the original articles, which cancer is described. For example, line 437 does not state the tumour type. I suggest indicating the tumour type or stating the studies concern melanoma without mentioning otherwise.

Subchapter 4.1.2. lines 187-212 should be introductory to CTLA-4 expression beyond T effector cells. However, the text focuses on specific aspects and largely on pDCs. I suggest separating the pDC in another paragraph or considering other re-organisation.

The paragraph 4.1.2.3 describing the role of B cells is very speculative. I don't think it provides a clear connection to human cancer. Especially the large part focusing on B1 cells, a specific B cell population that may be relevant to fewer cancer types.

The combined therapy nivolumab + ipilimumab is considered in the context of anti-PD-1/anti-PD-L1 (lines 322-333). Consider separating.

In Chapter 4.2.1.5. basophils only appear twice, in the title and first sentence, thus the title should be changed.

Chapter 4.2.1.8. In the text, APCs seem to be used as a synonym to DC which is incorrect. Other cells are known to act as APCs, also mentioned in lines 303-304. In line 625 "DC were also shown to express PD-L1" - it was mentioned in the introduction that PD-L1 is typically expressed in APC (lines 303-304). This quote shows uncertainty and does not connect to the previous text.

Author Response

Immune checkpoint inhibitors have marked the treatment of cancer. Their main role is blocking T-cell inhibitory signalling axis. However, the surface molecules aimed at by ICI are expressed by other immune and non-immune cells. The authors provide an overview of these cells, together with the evidence for their importance in response to ICI therapy. The review provides a rich resource to the literature investigating checkpoint inhibitor expression.

In chapter 2 a rather extensive description of melanoma is given. However, in the further text, other cancer types are mentioned. Sometimes it is not clear without going back to the original articles, which cancer is described. For example, line 437 does not state the tumour type. I suggest indicating the tumour type or stating the studies concern melanoma without mentioning otherwise.

Information on the according tumor types has been added.

Subchapter 4.1.2. lines 187-212 should be introductory to CTLA-4 expression beyond T effector cells. However, the text focuses on specific aspects and largely on pDCs. I suggest separating the pDC in another paragraph or considering other re-organisation.

The text section describing CTLA-4 induced IDO induction in DC has been transfered into the DC section (4.1.2.2.)).

The paragraph 4.1.2.3 describing the role of B cells is very speculative. I don't think it provides a clear connection to human cancer. Especially the large part focusing on B1 cells, a specific B cell population that may be relevant to fewer cancer types.We have shortened this paragraph and indicate thaht further studies are necessary to elaborate on the role of CTLA-4 on B cells with regard to ICI tumor therapy.

The combined therapy nivolumab + ipilimumab is considered in the context of anti-PD-1/anti-PD-L1 (lines 322-333). Consider separating.

We discuss in that paragraph (also) the outcome of nicolumab+ipilimumab treatment since this clinical trial has been the base for subsequent clinical approval of nivolumab.

In Chapter 4.2.1.5. basophils only appear twice, in the title and first sentence, thus the title should be changed.

We moved the paragraph and altered the headline (4.2.1.8. Other granulocytes).

Chapter 4.2.1.8. In the text, APCs seem to be used as a synonym to DC which is incorrect. Other cells are known to act as APCs, also mentioned in lines 303-304. In line 625 "DC were also shown to express PD-L1" - it was mentioned in the introduction that PD-L1 is typically expressed in APC (lines 303-304). This quote shows uncertainty and does not connect to the previous text.

Wording has been adjusted to clarify that APC and DC are not synonomous and to clarify in which cell type according findings have been made.

We thank the reviewer for his comments and suggestions.

Reviewer 3 Report

This review article addresses an interesting topic. In general, it nicely condenses a lot of findings in a well-structured manner. However, there are a number of slight inaccuracies, wording issues and ommisions that should be addressed in the course of a revision. To facilitate this process I am attaching an annotated pdf version of the original submission.

Author Response

Line 94: must be transforming growth factor

Mistake corrected.

Line 95: Among all known cytokines TGF-beta has the strongest immunosuppressive properties. Listing TGF-ß as a cytokine that would contribute to chronic inflammation within the TME is at least questionable.

TFGF-ß deleted from listing.

Line 106: T cells rarely express ligands for immune checkpoint receptors.

The statement has been revised.

Line 136: Exclusion of effector T cells and other cells that could orchestrate an anti-tumoral immune response is key for the survival of many tumors. While this is a crucial mechanism that would deserve more emphasis, inhibition of T cell infiltration does not happen within the TME, but often at the vascular barrier.

Statement altered/extended as suggested.

Line 200: one hypothesis is that ADCC-enhanced anti-CTLA-4 antibodies may label regulatory T cells which are then deleted. Therefore, ADCC-enhanced are being  explored in clinical trials. This hypothesis (which is strongly supported by work from Sergio Quezada) should therefore be mentioned.

We have extended this passage accordingly.

Line 246: ligation of CTLA-4 on B cells... The way this is written implies reverse signaling via CTLA-4. However, signaling via an intracellular CTLA-4 domain is unlikely to be a major mechanism. Instead, CTLA-4 on B cells may engage CD28 on T cells that would e.g. need to provide T cell help for isotype switching. Please suggest a plausible mechanism for the imhibitory role of CTLA-4 on B cells.

Line 271: The authors apparently use the term ligation  whenever there is an interaction. However, when there is a well-defined ligand-receptor interaction, it is confusing when they always ligate the ligand. In this context, ligation usually means that the receptor is ligated by its ligand. Thus, I would strongly recommend to use wordings like "binding of CTLA-4 on B cells presumably to CD80/86 on activated T cells", or "interaction of CTLA-4 on B cells presumably with CD80/86 on activated T cells". Please note that the text abounds with examples for this wording issue.

We have altered the wording as suggested.

Line 275: Wouldn´t it be much better to simply write "CTLA_4 expression has also been reported" rather than "Accounts of CTLA-4 expression have also been reported..."

The text has been altere das sugested.

Line 278: This is the next example for the overly frequent and injudicious use of the word "ligation".

Wording has been altered as suggested.

Line 285: This comes close to nonsense. First of all, CTLA-4 is not a receptor, but a ligand with a very well-defined immune inhibitory function. Further, chronic stimulation of activating receptors often leads to these receptors becoming desensitized which may be the case for CD28 in CTLA-4 haploinsufficient patients.

The questionable statement has been deleted.

Line 296: This is at least questionable. To the best of my knowledge this has not been reliably reproduced since its publication in 2005.

An according statement has been added.

Line 361: Some of the clinical approvals require stratification for patients with a certain percentage of PD-L1-positive tumor cells, or other PD-L1 positive cells in the TME. However, PD-L1 expression as assessed today is a very poor predictor for response. Could this also be due to a lack of attention towards other PD-L1 expressing cell types? Please discuss.

We discuss this issue.

Line 467: These indirect effects are obviously important for PD-1-based cancer immunotherapy, and they are also clearly described. However, for the purpose of this review article, it would be nice if the authors systematically distinguished between direct and indirect effect of immune checkpoint blockers on non-T cells.

We aimed to differentiate between direct/indirect effects throughout the manuscript.

Line 680: Ok. Still, as many immune cells are also present in NOD/SCID-mice, the effect may well  be immune-mediated. The fact that the PD-1/PD-L1 axis is relevant for many types of immune cells beyond T cells indicates that effects in the absence of T cells do not necessarily imply direct tumor cell-intrinsic, pro-proliferative effect on hepatocellular cancer cells. Indirect immune-mediated effects appear more likely. Nevertheless, there are reports on reverse signaling via PD-L1 in cancer cells. These would deserve to be mentioned.

Line 715: While this provides a nice and concise summary of emerging checkpoints, some information regarding the current state of clinical development for these ICB would be helpful for many readers (clinical phase, reported outcome).

Due to the vast number of (ongoing) clinical trials on emerging immune checkpoint inhbitors we refrained from addressing this broad issue in our review article. However, we had refered to a review article which focused on this issue (Lee et al., 2021) and have cited additional articles which discuss the outcome of clinical trials on new checkpint inhibitors in detail. Further, we briefly discuss that of these the LAG-3 blocking antibody relatlimab has been successfully tested in clinical trials and awaits approval for threapy of metastatic melanoma.

We thank the reviewer for his helpful comments.

Round 2

Reviewer 1 Report

The authors have addressed all my comments.

Author Response

Thank you

Reviewer 3 Report

While the authors have applied the requested changes, I still stumbled over two inaccuracies (lines 103 and 403) which ought to be corrected (and two editorial changes, lines 148 and 341). Please find my comments in the attached pdf.

Author Response

Please found in the attachment
